# Transthyretin Cardiac Amyloidosis: A Cardio-Orthopedic Disease

**DOI:** 10.3390/biomedicines10123226

**Published:** 2022-12-12

**Authors:** Federico Perfetto, Mattia Zampieri, Giulia Bandini, Roberto Fedi, Roberto Tarquini, Raffaella Santi, Luca Novelli, Marco Allinovi, Alessia Argirò, Francesco Cappelli

**Affiliations:** 1Tuscan Regional Amyloidosis Centre, Careggi University Hospital, 50134 Florence, Italy; 2Department of Experimental and Clinical Medicine, Careggi University Hospital, 50134 Florence, Italy; 3Division of Internal Medicine I, San Giuseppe Hospital, 50053 Empoli, Italy; 4Pathological Anatomy Unit, Careggi University Hospital, 50134 Florence, Italy

**Keywords:** transthyretin cardiac amyloidosis, musculoskeletal red flags, carpal tunnel syndrome, lumbar spine stenosis, early diagnosis

## Abstract

Orthopaedic manifestations of wild-type transthyretin amyloidosis are frequent and characteristic, including idiopathic bilateral carpal tunnel syndrome, idiopathic lumbar canal stenosis, atraumatic rupture of the brachial biceps tendon, and, more rarely, finger disease and rotator cuff. These manifestations often coexisting in the same patient, frequently male and aged, steadily precede cardiac involvement inducing a rapidly progressive heart failure with preserved ejection fraction. Although transthyretin cardiac amyloidosis remains a cardiac relevant disease, these extracardiac localisation may increase diagnostic suspicion and allow for early diagnosis assuming the role of useful diagnostic red flags, especially in light of new therapeutic opportunities that can slow or stop the progression of the disease. For the cardiologist, the recognition of these extracardiac red flags is of considerable importance to reinforce an otherwise less emerging diagnostic suspicion. For orthopedists and rheumatologists, the presence in an old patient with or without clinical manifestations of cardiovascular disease, of an unexpected and inexplicable constellation of musculoskeletal symptoms, can represent a fundamental moment for an early diagnosis and treatment is improving a patient’s outcome.

## 1. Introduction

Transthyretin cardiac amyloidosis (ATTR-CA) is a disease due to the extracellular myocardial deposit of misfolded transthyretin which causes structural alterations, dysfunctions, and early death. There are two distinct types of ATTR; a rare inherited genetic variant of TTR (ATTRv), and a more common type where amyloid fibrils are derived from the wild-type of TTR (ATTRwt), also known as senile systemic amyloidosis (SSA) to underline its specific link with the heart. Once the amyloid fibrils have heavily accumulated in the heart, their removal is virtually impossible and the patient experiences a rapid and irreversible deterioration in heart function with a median overall survival of 3.5 years [1]. Transthyretin (TTR) is a small homotetrameric protein synthesized mainly by the liver (over 90%) and released in the bloodstream where it acts as a thyroxine and holo-retinol carrier. The TTR molecule holds a natural structural instability which in some conditions, such as the incorporation of pathogenic mutations in the tetrameric form (ATTRv) or the tetrameric dissociation associated with aging (ATTRwt), accentuates the misfolding producing highly unstable intermediates and toxic compounds [2]. These compounds precipitate in the tissues as insoluble amyloid fibrils, adding mechanical damage to the toxic one. In the ATTRwt this amyloid-forming fibril mechanism appears particularly amplified in older males targeting above all the heart. Until recently, the diagnostic workup of ATTR–CA included a histopathology demonstration of amyloid fibrils after endomyocardial biopsy, but advances in nuclear imaging enabled easier and timely diagnosis and allow non-invasive diagnosis with high specificity [3]. This caused a massive increase year by year of new cases of ATTRwt-CA becoming the most frequent form of amyloidosis diagnosed today and is being increasingly recognised as a main cause of heart failure (HF) with preserved ejection fraction in the elderly [4,5,6,7]. However, despite recent diagnostic and therapeutic advances, the diagnosis of ATTR-CA remains late and even missed diagnosis [8,9]. Current therapeutic strategies for ATTR are designed to decrease TTR hepatic synthesis by the degradation of its RNA (patisiran and inotersen) or to stabilize the tetrameric form of TTR thus preventing its dissociation and misfolding (tafamidis or AG10) [10,11,12,13]. In the age of new therapeutic opportunities that can slow or stop disease progression, prompt diagnosis and early treatment are crucial. A significant proportion of patients with ATTR develop amyloid deposition not only in the heart but also in musculoskeletal soft tissues, including ligaments, tendons, and articular cartilage foreshadowing characteristic and unexpected clinical rheumatologic or orthopedic picture. Although ATTR-CA remains a cardiac relevant disease, these extracardiac localisations may increase diagnostic suspicion and allow for early diagnosis assuming the role of useful diagnostic red flags. Furthermore, these extracardiac localizations can precede cardiac infiltration by many years and represent, for a long time, the only clinical manifestation of ATTR amyloidosis (Figure 1). In this regard, recently, the summary of the Working Group on Myocardial and Pericardial Diseases includes, among the extracardiac alarm signals in patients with compatible cardiac imaging for CA amyloidosis, the presence of bilateral carpal tunnel syndrome and tendon rupture of the biceps. In these patients, these orthopedic manifestations are sufficient to trigger the diagnostic algorithm to obtain a definite diagnosis [14]. These findings raise the need for increased awareness of ATTR amyloidosis even among rheumatologists, head surgeons, and orthopedists, especially in patients with early signs of cardiac involvement. These patients, if early recognized and in a pre-HF phase, can be immediately referred to centres experienced in cardiac amyloidosis where a specific and effective anti-amyloid therapy can be started. The main purpose of this systematic review is to highlight the association between some common rheumatologic and orthopedic conditions and ATTR-CA by emphasizing their role as diagnostic red flags useful for early diagnosis. Data from the literature concerning the ATTR prevalence in some musculoskeletal pathology and the prevalence of amyloid deposits on surgical biopsy are reviewed and commented in separate paragraphs and summarized in Appendix A.

## 2. Carpal Tunnel Syndrome

Amyloid deposition in the tenosynovial tissues of the carpal tunnel may cause median nerve compression and carpal tunnel syndrome (CTS) characterized by pain, numbness, and tingling in the first three hand fingers and the radial side of the ring finger. CTS shows a peak incidence at 40–60 years of age with females twice as likely to be affected than males [15,16]. The prevalence of CTS in ATTRwt is very high reaching 54% of cases [17,18,19]. Notably, in 10% of tissue samples obtained from flexor tenosynovium and transverse carpal ligament during CTS decompression, fibrils amyloid has been found in 10% of patients with a minimum age of 50 years in men and 60 years in women [20]. The presence of CTS, in an old male patient, is a major red flag for ATTR-CA, being associated with a 12-fold increased risk of having ATTRwt-CA, which rises to 31-fold if bilateral anticipating overt cardiac involvement from 5 to 10 years [21]. Vianello et al. in a study population of 53 male patients with bilateral CTS (median age 73 years) found unexplained left ventricular hypertrophy in 6 (11%) individuals, 4 of whom underwent bone scan (two patients resulted positive for heart retention of bone tracer both classified as grade 2 on the Perugini scale) and negative for TTR gene mutations leading to the diagnosis of wild-type ATTR-CA. Since the high (33%) prevalence of ATTR-CA among bilateral CTS patients with unexplained left ventricular hypertrophy the authors concluded that screening for ATTR-CA in such a population appeared feasible and effective [22]. CTS is a common finding, although to a lesser extent, (24–30%) also in ATTRv, especially in those with cardiogenic mutation suggesting a likely pathogenic association between the two conditions [20,21,22,23,24]. The appearance of CTS may also be the first sign of clinical evolution in presymptomatic carriers of TTR pathogenic variants associated with predominant cardiac phenotype, allowing the start of cardiac investigations capable of revealing an incipient myocardial involvement [24]. (Figure 2). Porcari et al. reported that more than 30% of patients, with available cardiologic evaluation at the time of surgery, had echocardiographic findings of unexplained left ventricular hypertrophy associated with an electrocardiographic profile and a clinical history suggestive of CA [25]. These patients had higher all-cause mortality and higher new onset of heart failure or hospitalization at 3.8 and 2.4 years after surgery, even after adjustment for confounding factors such as age, gender, and renal insufficiency. In particular, men aged > 65 years and women > 70 years with a cardiovascular clinical profile at risk of amyloidosis, could represent a subgroup of patients to screen for the presence of amyloid in the surgically excised tissue and then refer to an amyloidosis center expert. In this concern, Ladefoged et al. evaluated the prevalence of ATTRwt in a group of patients aged 60 years or older who had a recent history (1–6 years) of idiopathic CTS surgery [26]. These patients, at the time of CTS surgery, were screened for the presence of typical electrocardiographic, echocardiographic and cardiac biomarkers red flags for ATTRwt. All the 67 red-flagged patients were referred to perform a diagnostic 99mTc-DPD (technetium-99m 3,3-diphosphono-1,2-propanodicarboxylic acid) scan, and among the 57 patients who accepted, the prevalence of ATTRwt was 8.3% (10 patients). These patients were predominantly asymptomatic and showed significantly less cardiac structural change and functional impairment than a control group of age, sex, and CTS surgery-matched patients with ATTRwt-CA. This approach emphasizes the importance of patients undergoing surgery for idiopathic CTS of a cardiologic screening based on simple ATTRwt-CA red flags looking for pre-symptomatic diagnoses with profound advantages in terms of therapeutic efficacy.

## 3. Brachial Biceps Tendon Rupture

Brachial biceps tendon rupture (BBTR) is uncommon in the general population. Clayton and Court-Brown reported an incidence of BBTR of 0.53/100,000 population with a male-to-female ratio of 3:1.1, aged 56 to 74 years [27]. The risk factors include older age, smoking, and shoulder overuse or heavy overhead activities. This typical finding of BBTR on physical examination is known as a Popeye sign which is caused by bulging of the biceps muscle belly after rupturing of the tendon with a descent of the muscle in the middle part of the arm becoming more obvious with contraction [28]. Figure 3 shows the presence of the Popeye sign (A) in a patient with ATTRwt with evident cardiac involvement as indicated by the high cardiac uptake at a 99mTc-hydroxyl-methylene-diphosphonate (HMDP) scintigraphy (B). In a study by Geller et al. out of 111 patients with ATTRwt, BBTR was present in 33% of patients involving the dominant limb in 95% of cases and bilateral in 24%. In the control group of 40 age-matched patients with non-amyloid aetiology of heart failure, BBTR was reported in only 1 patient [29]. In a recent study of our group, we systematically assessed the prevalence of BBTR in 168 patients with ATTR-CA (141 ATTRwt and 27 ATTRv-CA patients with cardiogenic mutation) and 81 patients with hypertrophic cardiomyopathy and Anderson Fabry disease (HCM/AFD). The presence of BBTR was compared to a cohort of 93 age-matched controls with no history of heart disease [30]. Unilateral BBTR was present in 44% of patients with ATTR-CA, 8% of controls, and 1% in HCM/AFD. Bilateral BBTR was reported in 15% of patients with ATTR-CA, in 1 control subject, and none of the HCM/AFD group. In one ATTR-CA patient, one HCM patient, and two control subjects, BBTR was related to major trauma. In all other patients, BBTR occurred spontaneously or after light exertion. In ATTR-CA, 20% of patients with BBTR were unaware of a tendon injury, which was diagnosed as a spontaneous rupture on physical examination.

## 4. Lumbar Spine Stenosis

The significance of ATTR deposits in the ligamentum flavum has been a matter of debate. Data from single-center studies have demonstrated that lumbar spine stenosis (LSS) can frequently be found in the history of patients with ATTR in both variant and wild-type forms, often associated with CTS, both anticipating several years after the onset of cardiomyopathy [19,31,32,33,34]. The presence of amyloidosis and its typing has also been systematically searched for in biopsies of common orthopedic surgeries regardless of the demonstration of systemic amyloidosis. Wininger et al. in a large systematic review of amyloid deposition in patients undergoing lumbar spinal decompression found the presence of ATTR on 64/157 (41%) surgical specimens [35]. In two different studies, the thickness of the ligamentum flavum was found to be significantly greater in patients with LSS and ATTR positive biopsy than in those with negative biopsy. Furthermore, the thickness in ATTR-positive patients increased with age [36,37]. Eldhagen et al.in an attempt to assess the presence of presymptomatic cardiac amyloidosis in a group of patients undergoing surgery for LSS and significant amounts of ATTR in the ligamentum flavum biopsies did not reveal any sign of cardiac amyloid involvement [38]. Nevertheless, the authors report the presence of five male patients with a history of CTS and a distribution and a typical amyloid fibrils pattern A (mixture of whole and truncated TTR fibrils) indistinguishable from that extracted in the systemic ATTRwt [38]. These data suggest that ATTR deposits in ligamentum flavum may precede further deposits at the systemic level, including the heart and that these patients should undergo careful clinical and instrumental follow-up in subsequent years. Recently Godara et al. extended these results through a multidisciplinary evaluation of 324 consecutive patients undergoing surgery for symptomatic LSS demonstrating the unequivocal presence of ATTRwt deposits in the ligamentum flavum in 43 (13%) patients; these patients were older and a higher prevalence of CTS than patients without ATTR deposits [39]. In addition, 37 of these patients received a 99mTc-pyrophosphate (99mTc-PYP) bone scan with 4 (11%) patients showing grade 2 or 3 myocardial uptake indicative of cardiac ATTR amyloidosis. Among the other 33 patients (78%) who showed equivocal scans (grade 1 or 0 myocardial uptake), the authors suggest a careful annual cardiac evaluation to intercept any clinical or instrumental changes that allow an early diagnosis of ATTRwt-CA.

## 5. Hip and Knee Osteoarthritis

Articular cartilage, especially in large load-bearing joints such as the hip and knee can be a target for the deposition of amyloid fibrils, especially that of TTR type but the systemic significance of these ATTR deposits has been a subject of controversy [40,41,42]. Takanashi et al. analyzing 322 samples from 232 consecutive patients (31 men and 201 women) undergoing arthroplasty or total knee joint replacement for osteoarthritis found significant TTR-related amyloid deposition in 26 specimens (8.1%) obtained from 21 patients [43]. Patients with positive TTR biopsies were older and were mostly female (5 men and 16 women) and none were screened for the presence of concomitant presence of ATTRwt-CA. To note all patients with known diagnoses of systemic AL or ATTR amyloidosis, myeloma, and monoclonal gammopathy were excluded from the study. Wininger et al. in their systematic review of 382 biopsies from patients without cardiac amyloidosis undergoing hip or knee arthroplasty, identified 98 amyloid-positive biopsies, all but one typed as TTR type (1 had AL amyloidosis) [35]. Recently, Rubin et al. compared the prevalence of total hip (THA) and knee arthroplasties (TKA) in 313 patients with cardiac amyloidosis (172 with ATTR-CA, 141 with AL-CA) and the general population, reported that arthroplasty occurs more frequently (5.6 and 3.2 fold respectively for THA and TKA) in ATTR-CA patients than AL-CA and controls. On average, arthroplasties anticipated ATTR cardiac disease by 7.2 years (8.4 years earlier for THA and 6.3 years earlier for TKA) [44]. All of these studies suggest a higher prevalence of TTR-related amyloid in the knee and hip joints of elderly (men) patients and also a higher prevalence in patients with ATTRwt-CA of THA and TKA, but whether or not this deposition is a part of a systemic disease that may precede or coincide with cardiac deposition remains an open question.

## 6. Trigger Finger

Degeneration and thickening of the proximal portion of the synovial sheath for the flexor tendons at the base of the finger or thumb can lead to a finger trigger where one or more fingers lock in a forced, bent, or straightened position. Patients with amyloidosis frequently refer to the presence of a trigger fingers phenomena but studies regarding its prevalence are lacking, above all in the ATTR subtype [45]. Cordiner-Lawrie et al. found amyloid deposition of unknown origin in 23% of surgical biopsies of patients aged over 46 years undergoing surgery for an idiopathic trigger finger [46]. Trigger finger associated with CTS and LSS has been reported as an initial manifestation of familial amyloid polyneuropathy (FAP) in a patient with TTR (Ile107Val) variant amyloidosis [47]. Trigger finger anticipated FAP by about 20 years. Interestingly familial study showed a high occurrence of trigger fingers in the family. Hara et al.l, in a retrospective series of patients with idiopathic finger tenosynovitis, found the presence of TTR amyloid deposition in the tenosynovium or flexor tendon biopsies in 65% of patients. On average, patients with amyloid deposition showed a higher number of digits affected compared with negative TTR deposition [48]. Recently Sood and Lipira conducted a retrospective study in a large cohort of patients undergoing digit-trigger release (TR) and/or carpal tunnel release (CTR) surgery with the aim of assessing the subsequent risk of amyloidosis diagnosis after surgery [49]. In the adjusted subdistribution hazard ratios (sHR), they report that patients undergoing TR or CTR had a risk of amyloidosis 4,8, and 10,2 fold higher than controls and that this risk was further increased mainly in patients with concurrent TR and CTR. There was no excess mortality among patients with TR and/or CTR compared to controls. The onset of cardiac amyloidosis in patients with TR and CTR occurred with a median of 3.1 years after surgery. These data suggest that trigger finger patients undergoing TR with concomitant CTS have an increased risk of incident amyloidosis in the following years of surgery during which they should undergo a careful annual cardiac evaluation.

## 7. Rotator Cuff Disease

Shoulders are a common site of involvement of haemodialysis-related amyloidosis due to the extremely high retention of circulating serum β2-microglobulin (β2-m) which causes β2-m to misfold and settle on the synovium, periarticular tissues and subchondral bone [50]. A further, albeit rare, shoulders involvement is found in AL amyloidosis in which progressive deposition of amyloid leads to pain and stiffness associated with progressive shoulder enlargement configuring the so-called shoulder-pad sign (Figure 4) considered as pathognomonic of AL [51,52]. Thanks to the studies carried out in patients with haemodialysis-related amyloidosis, substantial experience has been gained in the aspect of the tendons of the shoulder and particularly the rotator cuff tendon infiltrated by amyloid. Thickening of the supraspinatus tendon greater than 7 mm on sonography is thought indicative of amyloidosis of the shoulder. Other sonographic findings of amyloidosis include the thickening of the rotator cuff tendon, the thickening of the synovial sheath around the long head of the biceps tendon, and the thickening of the subdeltoid bursa (Figure 5) [53,54].

Studies about the prevalence of rotator cuff tears in ATTR cardiomyopathy are lacking. Takashio et al. reported a case of a patient with a long history of LSS, right rotator cuff tears, and bilateral CTS, that during preoperative evaluation for LS surgical decompression showed typical amyloidosis red flags at transthoracic echocardiogram, cardiac MRI and 99mTc-labelled pyrophosphate scintigraphy [55]. The ATTR-CA was confirmed by TTR amyloid deposition in the myocardium and surgically excised ligamentum flavum. Although the presence of amyloid in the rotator cuff has not been histologically demonstrated, the clinical history of this patient suggests extensive and advanced systemic amyloidosis with cardiac and joint involvement including the periarticular tissues of the shoulder.

In the aforementioned study on the prevalence of THA and TKA arthroplasties in 313 patients with CA, Rubin et al. reported also a high (9.9%) prevalence of rotator cuff repair in ATTR patients than in AL patients (5%) [44]. No data are available on the lag between rotator cuff repair and clinical diagnosis of cardiac ATTR amyloidosis. Sueyoshi T, et al. in their research of amyloid deposition in 111 orthopedic surgery specimens identified 47 (42.3%) amyloid-positive samples, 39 of which were typed as TTR derived from 5 rotator cuff tendon specimens [56].

## 8. Musculoskeletal versus Cardiac Involvement in TTRwt Amyloidosis: Two Sides of the Same Coin or Two Different Pathologies?

The presence of TTR-related amyloid fibrils in the cartilage, tendons, and ligaments of many patients with common rheumatologic/orthopedic manifestations has been established for many years. The orthopedist and rheumatologist must be aware of the existence of this clinical manifestation which can represent a sort of musculoskeletal TTR amyloidosis, capable of generating or complicating an underlying osteoarthritic process evolving towards clinical pictures and painful syndrome that require medical and surgical interventions. Furthermore, this cluster of manifestations, frequently present in the clinical history of patients with ATTR-CA, raises the question of whether they may represent two different manifestations of the same systemic infiltrative disease at different times of onset. A possible link between these two clinical manifestations may arise from a putative mechanism of proteolysis/fibrillogenesis of TTR tetramers in the extracellular compartment, probably fibrine derived. Several studies suggest that a proteolytic cut of TTR could be enhanced by increased mechanical forces such as shear stress, that occur in vivo into the heart and/or at the weight-bearing joints, or into tenosynovial tissues subjected to heavy gravitational loading combined with continuous repetitive vigorous movement, such as carpal tunnel o lumbar spine [57,58,59]. ATTR-CA is a progressive time-related infiltrative cardiac disease that leads to patient death due to refractory heart failure or arrhythmia [8,60]. In the era of new therapeutic opportunities that may slow or halt disease progression, early diagnosis and treatment are critical [10,11,12,13]. The majority of patients with ATTRwt-CA are diagnosed in the 8th decade when patients are affected by a greater cluster of comorbidities, such as arterial hypertension, diabetes, coronary artery disease, atrial fibrillation, pacemaker implantation, renal failure, and also cognitive deterioration. Moreover, results from the ATTR-ACT trial emphasized that the efficacy of tafamidis was greater in patients with mild heart failure symptoms and that its beneficial therapeutic effects emerged after approximately 18 months [13]. Thus, each year gained in the recognition of ATTR-CA could warranty not only an early diagnosis but also a greater therapeutic efficacy. Therefore, diagnostic red flags may play a pivotal role in clinical practice, raising the suspicion of amyloidosis and finally allowing to make a definitive and early diagnosis. In this context, rheumatologic and orthopedic extracardiac red flags are important opportunities for early diagnosis. As reported in Figure 6 for the cardiologist, the knowledge and awareness of these extracardiac musculoskeletal red flags can induce or reinforce a clinical suspicion of cardiac amyloidosis that was not initially taken into account or considered unlikely and to start the necessary diagnostic work-up favoring a timely and early diagnosis. At the same time for orthopedists hand surgeons and rheumatologists, this concomitance of musculoskeletal red flags, especially in an elderly male patient, may lead to suspecting the presence of ATTR-CA and thus referring patients to specialists with experience in amyloidosis. In the presence of a compatible amyloid clinical cardiac phenotype, an electrocardiogram, an echocardiogram, and a cardiac biomarkers evaluation, looking for typical warning signs for cardiac amyloidosis are strongly suggested [61]. In these patients, if referred for surgery, careful examination of the surgical specimen should be recommended in order to highlight the amyloid deposit [62]. Patients with a positive biopsy but without evidence of heart disease or patients not suitable for surgery should however be referred to a cardiologist expert in amyloidosis to closely follow the clinical course of these patients, as some of these patients may develop ATTR-CA.

## 9. Conclusive Remarks

Many studies reported the presence of amyloid deposits within ligaments, tendons, and articular cartilages in various orthopedic disorders. When amyloid deposits were subtyped, TTR-derived fibrils were observed much more commonly than other amyloid fibril. Musculoskeletal manifestations appear 5 to 15 years before overt cardiovascular signs and symptoms. Awareness of these extracardiac red flags by cardiologists and the knowledge by orthopedists of the possible association between some musculoskeletal ATTR clinical manifestations and myocardial infiltration may unmask a diagnosis of TTR amyloidosis and early refer these patients to a center with expertise in this disease. The mechanisms capable of determining the deposition of misfolded TTR fibrils in these different unrelated tissues are not yet clarified. Factors such as shear forces and tissue proteolytic have been claimed but the pathogenesis of the disease requires further studies.

## Figures and Tables

**Figure 1 biomedicines-10-03226-f001:**
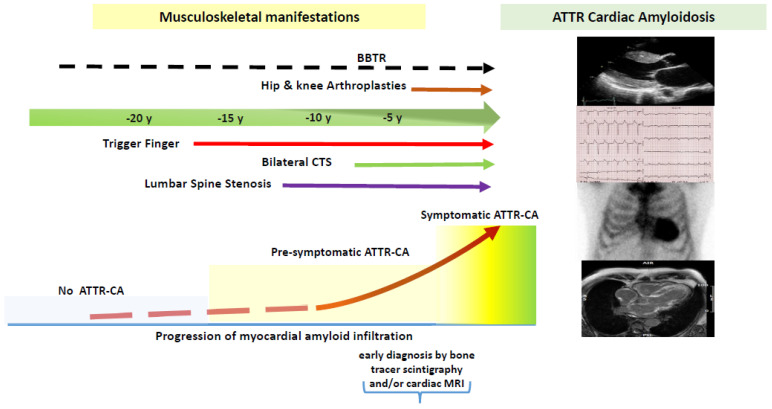
The Journey of ATTR Amyloidosis: the natural history of ATTRwt amyloid deposition from musculoskeletal tissues to myocardial infiltration. In the intermediate phase, symptomatic musculoskeletal infiltrations, coexisting with pre-symptomatic myocardial infiltration, can be useful clinical red flags allowing for an early diagnosis of cardiac amyloidosis through specific and sensitive tools such as bone tracer scintigraphy or cardiac MRI. BBTR: brachial biceps tendon rupture; TTR; Transthyretin; CA: cardiac amyloidosis; CTS: Carpal Tunnel Syndrome; MRI Magnetic resonance imaging.

**Figure 2 biomedicines-10-03226-f002:**
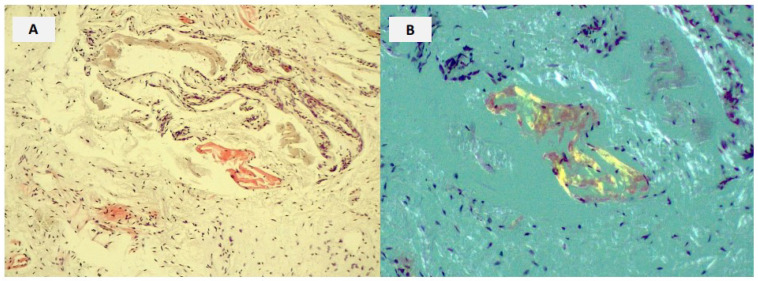
Histological examination of a tenosynovial surgical biopsy. Histological examination of a tenosynovial surgical biopsy from a 64-year-old male with bilateral CTS monitored at our institution because carrying an Ile68Leu variant TTR gene. The patient at the time of surgery was asymptomatic for heart failure. He was a brother of a patient who died 4 years before from an ATTR-CA Ile68Leu variant associated with bilateral STC. The presence of amyloid deposition is confirmed by the Congo red staining (**A**) with apple green birefringence under a polarized light microscope (**B**). Then the patient underwent a bone scan with 99mTc-hydroxyl-methylene-diphosphonate (HMDP) which showed moderate absorption of the myocardial bone tracer (Perugini grading score = 2) confirming the presence of TTR amyloid deposition in the heart although yet asymptomatic.

**Figure 3 biomedicines-10-03226-f003:**
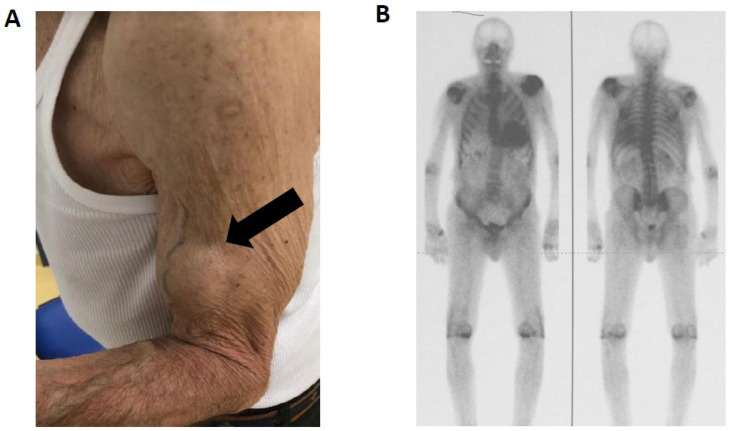
Popeye’s sign in an 82 year old patient wild-type transthyretin amyloidosis. (**A**) Popeye’s sign (Black Arrow). indicating rupture of the proximal biceps tendon. (**B**) In the same patient a 99mTc-hydroxyl-methylene-diphosphonate (HMDP) scintigraphy showed high myocardial uptake (grade 3 on Perugini Score).

**Figure 4 biomedicines-10-03226-f004:**
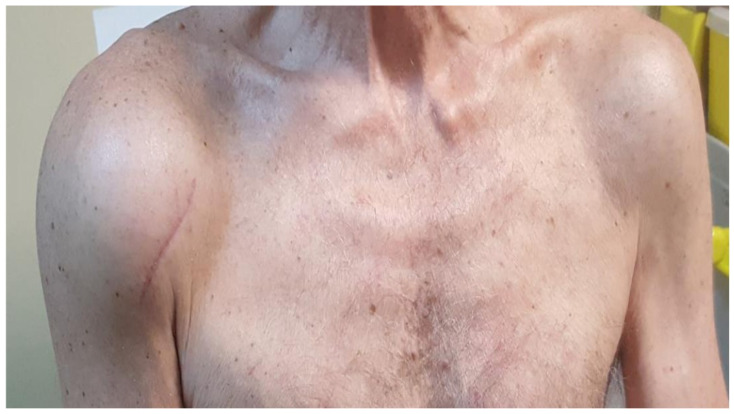
The shoulder pad sign. The shoulder pad sign, more evident on the right shoulder, determining pain and functional impotence of the left upper limb. The patient underwent a surgical biopsy of the shoulder (as evidenced by the residual surgical scar) which demonstrated the presence of extensive periarticular infiltration of amorphous, eosinophilic material, an intensely positive result for RC staining and green birefringence upon observation under an optical microscope with a polarizer. The patient also showed heart and peripheral nervous system involvement due to an IgGκ-type AL amyloidosis.

**Figure 5 biomedicines-10-03226-f005:**
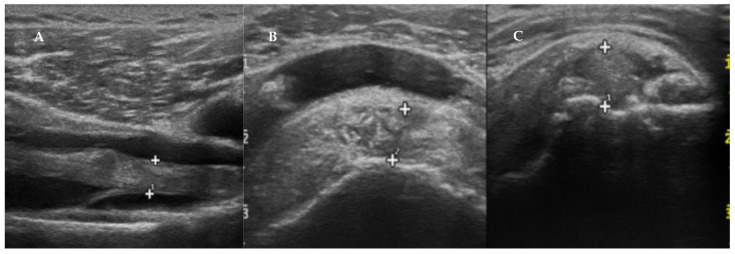
Sonographic findings of the shoulder of a patient with ATTRwt-CA. (**A**). Longitudinal sonography image of the long head of the biceps tendon showing a thickening of the tendon by hypoechoic, inhomogeneous material. (**B**,**C**). Longitudinal and transverse sonography images of the supraspinatus tendon show a non-homogeneous, thickened (8 mm) supraspinatus tendon with irregular echogenicity and subverted structure.

**Figure 6 biomedicines-10-03226-f006:**
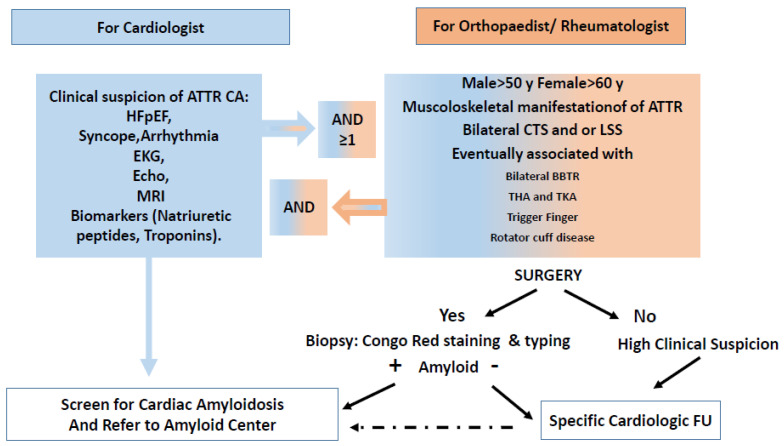
Clinical interconnection between the presence of musculoskeletal and cardiologic manifestations in the course of ATTR. For the cardiologist, the awareness of an orthopedic phenotype of TTR can induce or reinforce a clinical suspicion of cardiac amyloidosis otherwise considered unlikely while for the orthopedist hand surgeon or rheumatologist the presence of an unusual cluster of musculoskeletal manifestations in the same patient can induce a diagnostic suspicion of amyloidosis with activation of subsequent procedures of cardiologic screening or amyloid research in a surgical biopsy. In both cases, this increased clinical awareness can lead to an increase in early diagnosis of the disease. ATTR-CA. Transthyretin cardiac amyloidosis; HFpEF: heart failure with preserved ejection fraction; EKG: electrocardiogram, Echo: echocardiogram; MRI: Magnetic Resonance Imaging; CTS: carpal tunnel syndrome; BBTR: brachial biceps tendon rupture; LSS: lumbar spine stenosis, THA and TKA; total hip (H) and knee (K) arthroplasties. FU: follow up.

## Data Availability

Not applicable.

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
