# Peer review of "Transthyretin Cardiac Amyloidosis: A Cardio-Orthopedic Disease"

_biomedicines, 2022, doi:10.3390/biomedicines10123226_

Round 1

Reviewer 1 Report

The article entitled « transthyretin cardiac amyloidosis: more than a cardiologic disease” constitutes an important work of review of the literature on this subject. It provides a large amount of information by referencing 59 articles but the article requires some major modifications.

Only musculoskeletal abnormalities related to ATTR amylosis are analyzed, which is not explicitly announced in the title of the article. However, both cardiologists and rheumatologists will find useful knowledge to guide them towards the diagnosis of amyloidosis.

Our criticisms relating to this article mainly concern the form. The reading seems quite long and somewhat tedious because sentences are too long and there are frequent repetitions or redundancies (example lines 327-330) that would benefit from being lightened. As a result, the main take-home messages are somewhat buried in the text.

The title focuses on ATTR amyloidosis but it would be useful to specify (and to provide the corresponding reference) that osteo-articular lesions may also be linked to AL form despite a much weaker prevalence.

Many numeric values are reported from the literature, in the various rheumatic pathologies studied (carpal tunnel syndrome, brachial biceps tendon rupture, lumbar spin stenosis, hip/knee osteoarthritis and trigger finger), which makes the synthesis difficult for the reader. To enable the reader to better retain some useful orders of magnitude for practice, it would be interesting to draw up a table summarizing 1) the prevalence of osteo-articular clinical injuries in cases of recognized systemic amyloidosis and 2) the prevalence of amyloid deposits observed by biopsy in the presented clinical settings (knowing that there are variations between the different published works). In this regard, we would find it justified to better highlight the remarkable work of synthesis by Wininger et al (reference 35) who quantified these prevalences (reported only for lumbar spin stenosis and hip/knee osteoarthritis in this article ).

It is essential to reconsider figure calls and to provide figure legends (currently absent) that clearly explain what the reader needs to identify.

Call to figure 1 is inappropriate (line 66)

Call to figure 2 is inappropriate (line 106)

Call to figure 4A is inappropriate (line 136)

The second call to figure 2 (line 141) concerns the Popeye sign (coherent) but as for the other figures it is necessary to add a legend and explain figure 2B

Call to figure 4A is lacking

Figure 5 does not appear very clear owing to the arrows between the cardiologist side and the rheumatologist side (this may perhaps be less confusing with the help of a figure caption ?)

Section 8 is entitled ‘ATTRwt : the two side of the same coin or two different pathologies ?’ but you probably meant ‘Musculoskeletal vs cardiac ATTRwt involvement: the two sides of the same coin or two different pathologies ?’. Anyway, you do not answer the question.

The conclusion repeats verbatim the end of the summary, which must be changed. The main take-home messages could be 1) amyloid deposits are frequently observed in biopsy sample obtained during common osteo-articular diseases, 2) these findings may appear many years before the advent of systemic amyloidosis. 3) the mechanisms of filiation between these peripheral lesions and the systemic disease remain imperfectly understood. 4) Both the rheumatologist and cardiologist need to be aware of these red flags in order to make the diagnosis and begin treatment as soon as possible.

Some specific errors :

Reference 18 has no article title

Reference 22 is called before reference 20 (line 101)

Line 125 : these patients were…

line 142 finding (not find)

line 163: in these patients

line 248 author

line 346 EKG

Author Response

Thanks for the suggestions and the consequent modifications that have allowed a clear improvement of the manuscript

Question

1.Only musculoskeletal abnormalities related to ATTR amylosis are analyzed, which is not explicitly announced in the title of the article. However, both cardiologists and rheumatologists will find useful knowledge to guide them towards the diagnosis of amyloidosis.

Answer  As suggested by the Reviewer  the title of the aricle has been changed. The new title is:   Transthyretin cardiac amyloidosis: a cardio-orthopaedic disease

Question

  1. Our criticisms relating to this article mainly concern the form. The reading seems quite long and somewhat tedious because sentences are too long and there are frequent repetitions or redundancies (example lines 327-330) that would benefit from being lightened. As a result, the main take-home messages are somewhat buried in the text.

Answer Following the suggestions of the Reviewer some sentences have been reduced the length , others have been eliminated, others have been lightened. These changes have been made above all in chapter 8 trying to make it leaner and less repetitive. In addition, the title of the chapter has also been changed following the reviewer's suggestion. Thank you very much

Question

  1. The title focuses on ATTR amyloidosis but it would be useful to specify (and to provide the corresponding reference) that osteo-articular lesions may also be linked to AL form despite a much weaker prevalence.

Answer We understand the reviewer's aim to also include the musculoskeletal localizations of AL Amyloidosis in this review. However, in our clinical experience, these localizations are always negligible in the clinical picture of  AL  which is always dominated by cardiac, renal and  peripheral nervous system involvement. In most of our AL patients the articular manifestations are very subtle and not very useful in anticipating the diagnosis with the exception of the shoulder infiltration (the so called shoulder pad sign)  which however is quite rare. Even the association between STC and AL amyloidosis, much emphasized in the past, is no longer confirmed after the study by Rapezzi et all. (23)  In this study the prevalence of CTS was highest in ATTR patients with cardiac involvement (20.3%, 95% CI 15.3–26.0 vs. 4.1% in the general population) while it was comparable to the general population in AL patients.

  1. Milandri A, Farioli A, Gagliardi C, Longhi S, Salvi F, Curti S, Foffi S, Caponetti AG, Lorenzini M, Ferlini A, Rimessi P, Mattioli S, Violante FS, Rapezzi C. Carpal tunnel syndrome in cardiac amyloidosis: implications for early diagnosis and prognostic role across the spectrum of aetiologies. Eur J Heart Fail. 2020 Mar;22(3):507-515. doi: 10.1002/ejhf.1742. Epub 2020 Jan 23. PMID: 31975495.

Question

  1. Many numeric values are reported from the literature, in the various rheumatic pathologies studied (carpal tunnel syndrome, brachial biceps tendon rupture, lumbar spin stenosis, hip/knee osteoarthritis and trigger finger), which makes the synthesis difficult for the reader. To enable the reader to better retain some useful orders of magnitude for practice, it would be interesting to draw up a table summarizing 1) the prevalence of osteo-articular clinical injuries in cases of recognized systemic amyloidosis and 2) the prevalence of amyloid deposits observed by biopsy in the presented clinical settings (knowing that there are variations between the different published works). In this regard, we would find it justified to better highlight the remarkable work of synthesis by Wininger et al (reference 35) who quantified these prevalences (reported only for lumbar spin stenosis and hip/knee osteoarthritis in this article ).

Answer. As suggested by the reviewer, we drawn up a table (named Table 1) recalling the orthopaedic pathologies object to review in the article, summarizing next to the specific reference, the number of patients/biopsies enrolled, the average age of the patients studied (when known), the biopsies analysed and the main key findings. This made it possible to eliminate some sentences from the text, making it a little manageable.  Case report are not included

Question

It is essential to reconsider figure calls and to provide figure legends (currently absent) that clearly explain what the reader needs to identify.

Call to figure 1 is inappropriate (line 66)

Call to figure 2 is inappropriate (line 106)

Call to figure 4A is inappropriate (line 136)

The second call to figure 2 (line 141) concerns the Popeye sign (coherent) but as for the other figures it is necessary to add a legend and explain figure 2B

Call to figure 4A is lacking

Figure 5 does not appear very clear owing to the arrows between the cardiologist side and the rheumatologist side (this may perhaps be less confusing with the help of a figure caption ?)

Answer. The lack of figure legends is my fault alone. I had not included the separate file with the legends among the files to be sent to the Journal Now the legends  are put in  the end of the manuscript. Please note that figure 1 (central illustration. The Journey of TTR amyloidosis) was not included in the text. I left it among the files of the figures so that it can be introduced according to layout needs

Question

Section 8 is entitled ‘ATTRwt : the two side of the same coin or two different pathologies ?’ but you probably meant ‘Musculoskeletal vs cardiac ATTRwt involvement: the two sides of the same coin or two different pathologies ?’. Anyway, you do not answer the question.

Answer.  The title of the chapter has also been changed following the reviewer's suggestion. Thank you very much We have tried to answer the question about the title of the paragraph, reporting some studies  by the Vittorio Bellotti and the National Amyloidosis Center of London. These articles indicated  with the references 57, 58, 59 report a possible hypothesis of mechano-enzymatic cleavage of transthyretin as a pathogenetic mechanism of amyloidogenesis in ATTRwt. This mechanism appears to be particularly present in high shear stress tissues such as the heart and/or high gravitational load such as weight-bearing joints.

Some specific errors :

Reference 18 has no article title                                                      

  Reference 18 has been replaced (also the N° 17)

Reference 22 is called before reference 20 (line 101)               

  Done

Line 125 : these patients were…                                                       

Done. Thank you

line 142 finding (not find)                                                                   

Done. Thank you

line 163: in these patients                                                                  

Done. Thank you

line 248 author                                                                                     

Done. Thank you

line 346 EKG                                                                                          

we changes EKG with electrocardiogram in order to reduce the abbreviation as requested by Reviewer 2

Reviewer 2 Report

This is a very interesting review underlining the association of  othopaedic and rhematological disorders with transthyretine cardiac amyloidosis (ATTR). The recognition of these diseases should be an alert to perform further close cardiological follow-up, diagnose ATTR at an early stage and introduce effective treatment, which is now available. 

I have some minor remarks:

1) there are many abbreviations, probably too many; in any case they should be explained when first mentioned (which is not always the case - line 30-32 page 1). I suggest at least a side-table with the abbreviations

2) the sentences are often too long: page 8 line 330-338; page 9 line 369-376

3) the English language should be corrected: spelling, missing verbs

4) the figures should be properly subscribed / explained

Author Response

This is a very interesting review underlining the association of  othopaedic and rhematological disorders with transthyretine cardiac amyloidosis (ATTR). The recognition of these diseases should be an alert to perform further close cardiological follow-up, diagnose ATTR at an early stage and introduce effective treatment, which is now available

Thank you very much for appreciating our work

1) there are many abbreviations, probably too many; in any case they should be explained when first mentioned (which is not always the case - line 30-32 page I suggest at least a side-table with the abbreviations

According to  the Reviewer we add a side-table with the several abbreviations

2 the sentences are often too long: page 8 line 330-338; page 9 line 369-376

Following the suggestions of the Reviewers some sentences have been reduced the length , others have been eliminated, others have been lightened. These changes have been made above all in chapter 8 trying to make it slender and less repetitive. 

3) the English language should be corrected: spelling, missing verbs

We have undergone a major text revision to improve the English language and the errors have been corrected. Thank you

4) the figures should be properly subscribed / explained

.The lack of figure legends is my fault alone. I had not included the separate file with the legends among the files to be sent to the Journal.  Now the legends  are put in the end of the manuscript. Please note that figure 1 (central illustration. The Journey of TTR amyloidosis) was not included in the text. I left it among the files of the figures so that it can be introduced according to layout needs

Round 2

Reviewer 1 Report

Decembre 1st 2022
Comments on the revised article: Transthyretin cardiac amyloidosis: a cardio-orthopedic disease:
The author has taken all the comments into account and has significantly modified his article in a convincing manner.
We point out here some remaining remarks, minor, requiring a final editing
Several small corrections have been indicated in the text (highlighted in yellow), see biomedicines-2043462-peer-review-v2-re-read.pdf
Otherwise:
The terminology should be standardized because it is sometimes written TTR-CA and sometimes ATTR-CA
Figure 1 caption, apart from the very long sentences, making it quite difficult to read, I'm not sure that the last sentence is necessary here “The majority of patients were unaware of the onset or existence of BBRT.“,  because page 5 line 179 is written “In ATTR-CA, 20% of patients with BBTR were unaware of a tendon injury”
Figure 2: letters A and B are lacking on the figure
Figure 3 are A and B from the same patient ? no call for fig 3B (in contrast to fig 3A)
Figure 4 letters A, B C lacking
Figure 6 syncope not sincope
Finally, it seems that the format of the bibliographical references is not always in accordance with the rules of the journal

Author Response

Thank you for revision and suggestions.

Here below my corrections point by point

  1. The terminology should be standardized because it is sometimes written TTR-CA and sometimes ATTR-CA

Done including in Figure 1.

  1. Figure 1 caption: the caption was shortened and rephrased as suggested
  2. Figure 2: letters A and B are lacking on the figure

Done

4A. Figure 3 are A and B from the same patient ?

Yes

 4B. no call for fig 3B (in contrast to fig 3A)

 The following sentence has been added to the text “Figure 3 shows the presence of the Popey sign (A) in a patient with ATTRwt with evident cardiac involvement as indicated by the high cardiac uptake at a 99mTc-hydroxyl-methylene-diphosphonate (HMDP) scintigraphy (B).”

  1. Figure 4 letters A, B C lacking

Done

  1. Figure 6 syncope not sincope

Done

7 it seems that the format of the bibliographical references is not always in accordance with the rules of the journal

Done